# Status of human onchocerciasis transmission in the Adamaoua region of Cameroon after 20 years of ivermectin mass distribution

Philippe Bienvenu Nwane[1,2]*, Hugues Clotaire Nana-Djeunga[1], Narcisse Nzune Toche[1], André Domché[1,2], Fesuh Nono Bertrand[3], Yannick Emalio Niamsi[1], Guy Roger Njitchuang[1], Martine Augusta Flore Tsasse[1], Jean Bopda[1], Steve Mbickmen[1], Aubin Balog[1], Alexis Nkwelle[4], Patrice Nkwelle[4†], Clarisse Ebene[5], Honoré Obama[5], Paul Messi[5], Benjamin Biholong[5], Serge Billong[5], Georges Nko'o Ayissi[6], Joseph Kamgno[1,7]

**1** Higher Institute of Scientific and Medical Research (ISM), Yaoundé, Cameroon, **2** Laboratory of General Biology, University of Yaoundé I, Yaoundé, Cameroon, **3** National Advanced School of Engineering, University of Yaoundé I, Yaoundé, Cameroon, **4** International Eye Foundation, Yaoundé, Cameroon, **5** National Onchocerciasis Control Program/Ministry of Public Health, Yaounde, Cameroon, **6** Department of Neglected Tropical Diseases, Ministry of Public Health, Yaounde, Cameroon, **7** Faculty of Medicine and Biomedical Sciences, University of Yaounde I, Yaounde, Cameroon

† Deceased.
* philino07@yahoo.fr

## Abstract

### Introduction

Significant progress has been made in onchocerciasis control through mass distribution of ivermectin among affected human populations, fostering optimism for disease elimination. However, despite these considerable advances, the elimination of the disease remains a major challenge in many African foci. This paper describes the current situation of onchocerciasis in Adamaoua Region of Cameroon after 20 consecutive years of ivermectin mass treatment.

### Materials and methods

The study was conducted between August and September 2020 in Adamaoua Region of Cameroon. Onchocerciasis endemicity was assessed through parasitological and clinical diagnosis. Microfilarodermia and nodule prevalences assessed in 2020 were compared to those of 1998-2002 and 2010-2013 surveys using the *Chi-square* ($X^2$) statistic test.

### Results

A total of 4,814 participants aged between 5 and 108 years, including 50.4% men and 49.6% women were enrolled in the study. The nodule and microfilaria prevalences reported from this sub-sample were 0.87 [0.64 - 1.19] % and 0.77 [0.54 - 1.07] %, respectively. At the community level, the mf prevalences ranged from 0.5% to 4.5%. Globally, the community microfilarial loads (CMFL) were < 0.5 mf/ss. The survey therapeutic coverage rates were between 40% and 78%, lower than those reported (79% - 83%) by the NOCP.

**Data availability statement:** All data are in the manuscript and/or supporting information files.

**Funding:** This study was funded by the Kreditanstalt für Wiederaufbau (kwF), through the Project for Control of Neglected Tropical Diseases in Central Africa (MTN/OCEAC) Internal BMZ N° 2015. 69.227+ 2016. 68797 to PBN. This funding has been allocated to the NGDO International Eye Foundation, which supports the Ministry of Public Health in the fight against onchocerciasis in the Adamaoua Region. The funders had no role in study design, data collection, data analysis, data interpretation, writing of the report, or the decision to submit for publication.

**Competing interests:** The authors have declared that no competing interests exist.

The coverage rates in ivermectin treatment in all age groups of the population were below 65%, except for the 40-50 age group where it was ≈70%.

## Conclusion

The results of this study show a drastic decline in onchocerciasis prevalences after 20 consecutive years of CDTI, indicating a significant progress towards stopping *O. volvulus* transmission in Adamaoua Region. However, additional efforts are needed to increase the population coverage in ivermectin treatment in order to stop the parasite transmission in this region.

### Author summary

Onchocerciasis control is mainly based on mass treatment of populations with an annual dose of ivermectin. At the early implementation stage of onchocerciasis control activities, it was suggested that an onchocerciasis-endemic community using the community-directed treatment with ivermectin (CDTI) approach should become free of the disease after 15 years, a period corresponding to the maximum lifespan of the parasite adult worm in humans. CDTI contributed to eliminate onchocerciasis in few foci in East and West African countries (e.g.,: Uganda, Mali, Niger, Senegal and Nigeria). In Central Africa, the elimination of this disease remains uncertain after 20 consecutive years of mass treatment in certain foci. The data presented in this study are very promising and suggest that the CDTI approach may stop in the short term the transmission of *O. volvulus* in the Adamaoua Region. In order to achieve the objective of eliminating onchocerciasis in the near future in this part of the country, efforts of population to comply with ivermectin treatment are essential. This will involve active raising awareness among men and women of all eligible age groups of the population in Adamaoua Region.

## Introduction

Also known as "*River Blindness*" onchocerciasis is caused by a parasitic worm *Onchocerca volvulus* transmitted from person to person through bites of female blackflies belonging to the genus *Simulium*. Over the course of few months to one year, the transmitted $L_3$ infective stages of the parasite molt twice and develop into sexually mature adult. Female worms live in fibrous nodules of the skin where they remain sessile, while males migrate between the nodules to mate with settled females. Both female and male adult worms live between 2 and 15 years with an average life expectancy of 10 years [1,2]. During this period, patent females release between 700 and 1,500 microfilariae (mf) per day [1,3], which predominantly live in the skin for 3 to 5 years causing lesions including intense itching, thickening, skin depigmentation, and loss of its elasticity. Other clinical manifestations such as hanging groin, skin atrophy, papular onchodermatitis and chronic papular onchodermatitis, lichenified onchodermatitis are commonly reported [4]. The microfilaria (mf) can migrate throughout the intercellular fluid and reach the eyes causing ocular manifestations which can lead to irreversible blindness [5]. Although the physiopathology is not well known, several studies have mentioned links between onchocerciasis, epilepsy and nodding syndrome [3,6–10]. Onchocerciasis occurs in 35 countries in the world with Sub-Sahara African (SSA) countries carrying over 99% of all cases in the world [11–14]. Historical data from SSA show a bleak picture of

this disease in several endemic villages where prevalences of infected individuals sometimes exceeded 80%, and the rates of onchocercal blindness around 10% [15–17]. In regard with the gravity of this severe infectious disease, private, public, national and international health authorities/agencies have decided to reduce its prevalence and transmission, then its elimination in affected countries [18,19]. Since then, the control of the disease has been based on 2 major large-scale strategies including (1) vector control targeting *Simulium* blackflies and (2) parasite control targeting *O. volvulus* in humans. These two strategies have been implemented at large scale in SSA to alleviate the burden of the disease in human populations. The first strategy was implemented in West Africa between 1974 - 2002 as a core activity of Onchocerciasis Control Programme (OCP) [15]. The second strategy targeting parasite control was proposed with the discovery of a safe and effective microfilaricide, ivermectin known under the brand name Mectizan [20]. Since 1987 this drug is freely provided by the pharmaceutical company Merck & Co [21]. The availability of ivermectin in sufficient quantities has enabled to continue the control of onchocerciasis in West Africa and its extension in endemic countries of Central and East Africa under the management of African Programme for Onchocerciasis Control (APOC) [22,23]. The APOC adopted Community-Directed Treatment with Ivermectin (CDTI) as a core strategy with the objective to establish a mechanism for sustained delivery of an annual dose of ivermectin to the entire eligible populations in meso- and hyperendemic communities [24,25]. Based on the lifespan of adult worms in humans, the expectation of eliminating onchocerciasis after 15 consecutive years was a convincing idea for testing vector control in West Africa and implementing CDTI in all endemic countries in SSA [26].

However, in countries where vector control or CDTI has been successful, the duration of implementation of each strategy was above 15 years, slightly longer than the lifespan of *O. volvulus* adult worm in human [27]. Evidences of onchocerciasis elimination through CDTI approach in some foci of Latin America and SSA [28–35] have supported the idea of shifting from the control to elimination of the disease [36]. Besides, the evidences, it is increasingly recognized that onchocerciasis is no longer a public health or socio-economic concern after 15 to 20 successive years of CDTI in some foci in West Africa [37]. In Central Africa, such a statement remains anecdotal after 20 years of CDTI, without any hope or evidence of eliminating the disease.

The 1998-2002 REMO surveys conducted in Cameroon resulted in the eligibility of the Adamaoua Region for ivermectin treatment. The CDTI was launched in 1999 in a part of the region (Adamaoua II) and was extended to the whole region in 2003. Epidemiological assessments conducted after the CDTI implementation in some endemic zones in Cameroon demonstrated that the transmission of onchocerciasis is still significantly ongoing even in foci where CDTI has been implemented for more than 15 years [35,38,39]. Those conducted by APOC in 2010-2013 in 7 CDTI projects in Cameroon showed that elimination of onchocerciasis was feasible in the short term in the Adamaoua II based on the decline in microfilarial prevalences noted in this part of the region [37]. Since then, the Adamaoua Region, like all the other regions in Cameroon has been receiving mass treatments of ivermectin every year. Apart from this 2010-2013 epidemiological survey conducted in 9 villages in Adamaoua II, no other survey covering the entire region has ever been carried out. This paper reports the situation of onchocerciasis prevalence in the Adamaoua Region after 20 consecutive years of mass treatment of populations with ivermectin.

## Materials and methods

### Ethics statement

A written informed consent was obtained for all participants aged 18 and above. Because the age of legal majority in Cameroon is 18, a written consent was obtained from the

parent/guardian for participants aged 5 to 17 years. Each study participant aged 18 and above personally affixed both thumbprints to the written informed consent form presented to them to confirm their participation in the study The study protocol was approved by the Cameroon National Ethics Committee (N°2020/08/1286/CE/CNERSH/SP), on 21 August 2020.

## Study area

This study was conducted in the Adamaoua Region of Cameroon located between 7°20'N and 13°01'E. The region covers a surface area of about 62,000 km², with altitude ranging between 900 m and 1500 m. Geographically, Adamaoua Region is situated in the transition zone between the humid equatorial climate of the South and the tropical dry sudano-sahelian climate of the North, influenced by the Adamaoua Plateau [40]. The rainy season extends from April to October with an annual rainfall of 1,500 mm [41]. The dry season occurs between November to March with a mean annual temperature of 20-26°C [42]. Most of the watercourses in Cameroon have their source in the Adamaoua Region, so called the "Water-shep of Cameroon". The main rivers of the region include, Mayo Deo, Mbéré and Vina. These rivers are of a great importance for human activities such as agriculture and livestock breeding [43,44]. In their watercourse, they exhibit water patterns including rapids, cascades and waterfalls that are favorable to the establishment of *Simulium* blackfly populations. The health map divides the region in 9 health districts (HDs) including Ngaoundéré Rural, Ngaoundéré Urbain, Bankim, Tibati, Banyo, Ngaoundal, Tignere, Djohong and Meiganga. Onchocerciasis is known to be endemic in the Adamaoua Region for around 10 decades [45]. A total of 29 communities spread across the 9 HDs were selected, with a priority given to those visited during the 1998-2002 Rapid Epidemiological Mapping of Onchocerciasis (REMO) conducted in the region.

## Study design

A cross-sectional survey was conducted between August and September 2020. Data were collected from people living in several communities in the HDs of the Adamaoua Region. In these communities, the collection of biological samples was performed in public places particularly in local health facilities or to a lesser extent in schools for communities without health facilities. In each HD, 2 to 4 communities were prospected in the framework of this study. The selection of the surveyed communities was based on (1) the existence for the community a baseline parasitological data on onchocerciasis from the 1998-2002 REMO studies (2) the high density of human population in certain communities and (3) the location of communities near watercourses with potential *Simulium* breeding sites. The participation in this study by community members was voluntary. However, the inclusion criteria were as follows: (1) be at least 5 years old, (2) reside continuously in the community, (3) for adults, sign a written informed consent form during the registration and assent form for children. Because the age of legal majority in Cameroon is 18, a written consent was obtained from the parent/guardian for participants aged 5 to 17 years.The study participants' itinerary consisted of 3 stations namely registration, skin biopsy collection and general consultation (**Fig 1**). At the registration station, all adult participants provided written informed consent. Personal information of each participant (gender, age, occupation, permanent residence,) was recorded and a barcode number was assigned to each eligible participant to keep track of the data. At the skin biopsy collection station, 2 superficial skin biopsies were collected from each participant at the left and right posterior iliac crests using a sharp and sterilized 2-mm Holthtype corneoscleral punch. To avoid transmission of potential pathogens, one corneoscleral punch was used per

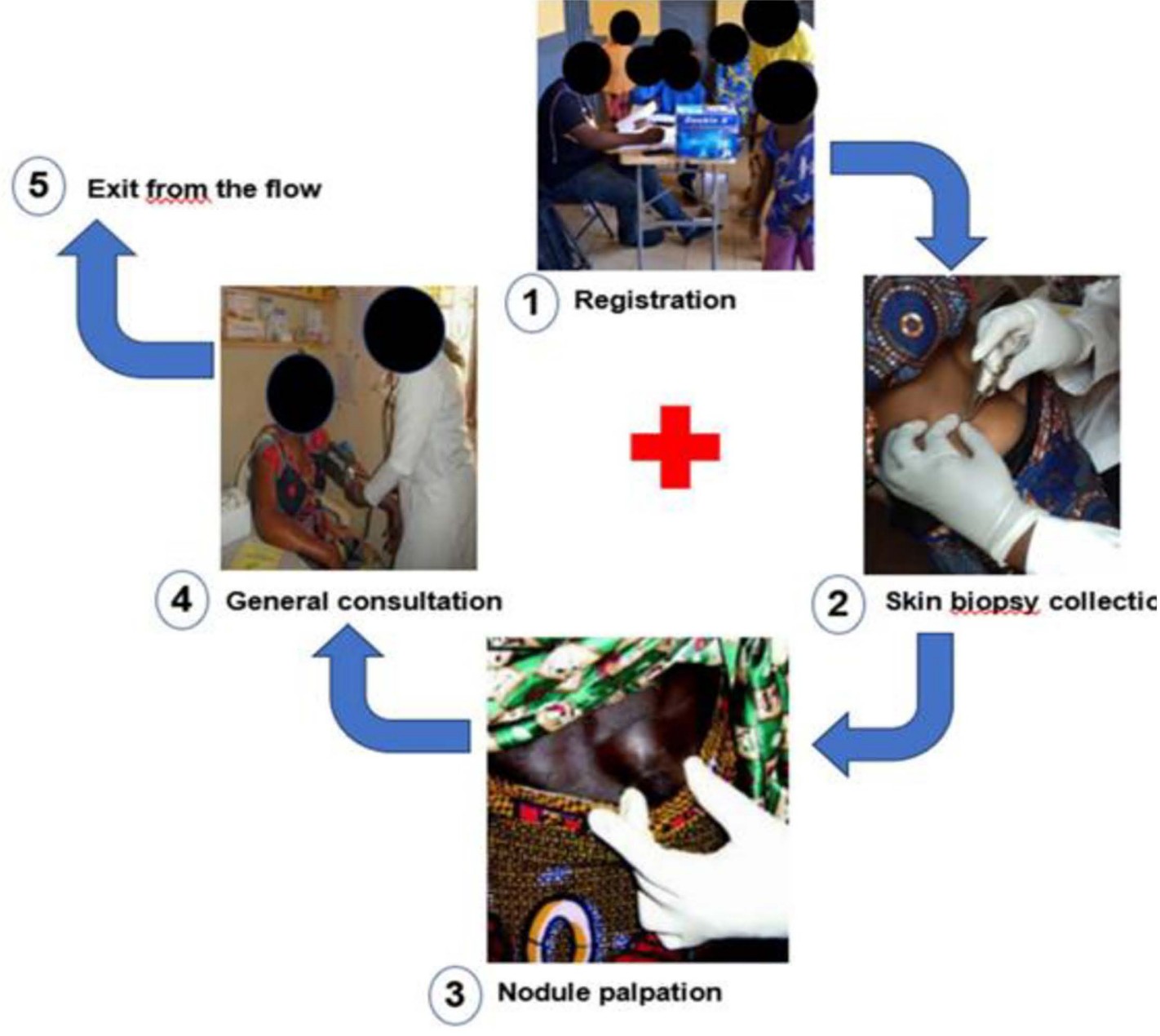

Figure 1. Participant progress flow during data collection procedure

➕ : Health facility ; (1), (2), (3), (4) et (5) : Registration/Examination stations

**Fig 1. Participant progress flow during data collection procedure.**

participant for the skin biopsy collection. After use, the corneoscleral punches were subjected to a complete sterilization procedure including 2 stages: (1) immersion for 10 to 15 minutes in bleach solutions of decreasing concentrations (6, 4 and 2 drops in 100 ml of distilled water) and (2) heating at 120°C for 30 minutes in a portable steam sterilization.

Immediately after collection, each skin biopsy was individually placed into a well of a round-bottom 96-well microtiter plate containing approximately 40μL of physiological saline per well. Plates containing skin biopsies were incubated for 24 hours at room temperature to induce microfilaria emergence [46]. At the end of this period, they were transferred to the local field laboratory for skin biopsy sample examination. In the laboratory, the physiological saline contained in each well was pipetted and placed on a slide for observation under a 40x magnification in order to search for and count the microfilariae. At the medical consultation station, the participant was subjected to onchocerciasis nodule palpation which consists of searching subcutaneous nodules over the body and the skin manifestations of the disease. The participant was then interviewed about his ivermectin treatment history. The answers provided were noted on an individual sheet that was submitted to the laboratory for archiving result of skin biopsy examination.

## Data analysis

Personal information, nodule palpation and skin snip biopsy data for each participant were recorded in a specially prepared spreadsheet. The main indicators of onchocerciasis including microfilaria (mf) and nodule prevalences were expressed as a percentage, i.e., number of persons positive divided by the number of persons examined × 100. The arithmetic mean of the mf from the two skin biopsies from each participant was calculated and used as a measure for intensity of infection. The community microfilarial load (CMFL) was calculated as anti-log $\{(\sum\log(x+1))/n\}$ -1, with $x$ being the mean of mf/mg (microfilaria per milligram) of skin and $n$ the number of individuals examined [47]. In the framework of this study, the methods described by Coffeng and colleagues [48] were used to convert nodule prevalences of the 1999-2001 survey (typically in samples of adult males aged ≥ 20 years) into microfilarial prevalences (in the population aged ≥ 5 years).

The chi-square ($\chi^2$) test was used to compare: (1) the microfilaria and nodule prevalences reported in this study with those of 1998-2002 (baseline) and 2010-2013 (APOC evaluation), (2) and the reported and surveyed coverage rates recorded in 2019 in the HDs of Adamaoua Region. A two-tailed p-value lower than 0.05 was considered statistically significant.

## Results

### Demographic variables

A total of 4,814 participants from 29 communities selected in the 9 HDs of the Adamaoua Region were enrolled in this survey. Men represented 50.4% (N = 2,424) and women 49.6% (N = 2,390). Participants were aged between 5 and 108 years old, with an average age of 31 years. The distribution of participants according to the study population by age group and gender is shown in **Fig 2**. Participants aged between 5 and 30 years represented more than a half of the study population (58.45%).

### Status of *O. volvulus* infection in the surveyed communities and health districts

Of the 29 communities surveyed during this study, 11 (≈ 38%) from 7 HDs were infested by *O. volvulus* microfilariae (**Table 1**). Of these, 4 were from Meiganga HD, 2 from Ngaoundal HD and 1 in each of the following HDs: Bankim, Djohong, Ngaoundéré Rural, Ngaoundéré Urbain and Tibati. No infected community was found Banyo and Tignere HDs. The mf prevalence among infected communities ranged from 0.50% [0.02 - 3.23] in Mbakaou (Tibati HD) to 4.34% [1.77 - 9.63] in Nandeke (Meiganga HD). These mf prevalences were significantly lower

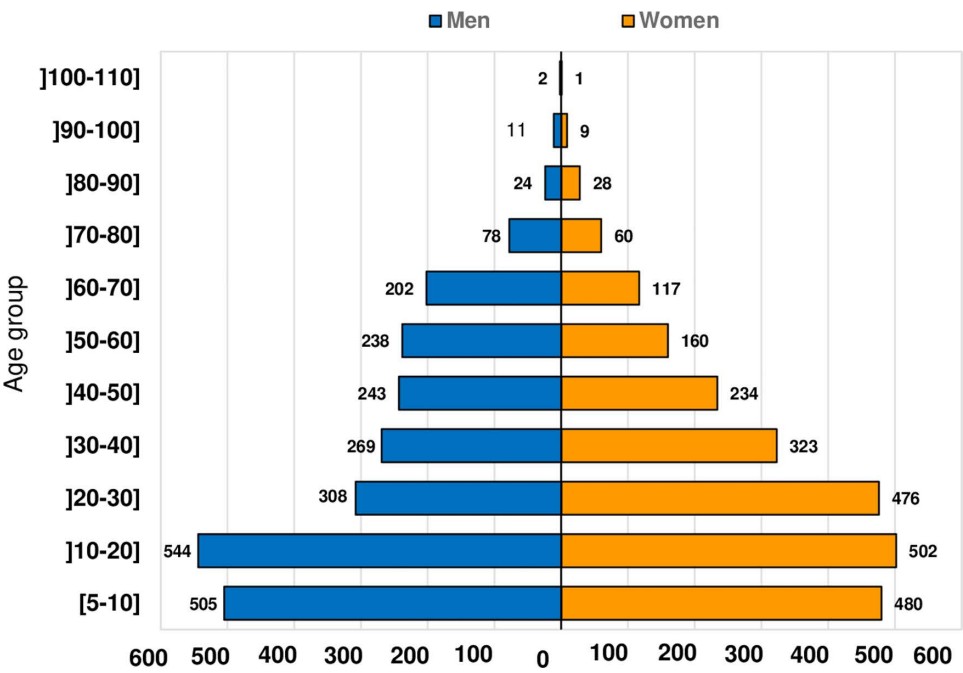

**Fig 2. Distribution of the study population according to gender and age groups.**

(p < 0.05%) compared to those recorded in the 1998-2002 baseline survey which varied from 36.7% in Alhamissa (Tibati HD) to 87.7% in Djamtari (Tignere HD). Of the 29 communities visited in the 2020 follow-up survey, 22 were previously prospected in the 1998-2002 baseline survey. The variation trend in mf prevalences in these 22 communities from the 1998-2002 baseline survey to the 2020 follow-up (after almost 20 years) is shown in **Fig 3**. Of these 22 communities, 13 (59%) were found free of *O. volvulus* infection and 9 (41%) showed mf prevalences below 5% (**Table 1**) in the 2020 study. In general, the intensity of *O. volvulus* infection characterized by the community microfilarial load (CMFL) was very low (< 0.5 mf/ss) in each of the surveyed communities. The CMFLs assessed in the 2020 survey range from 0.006 mf/ss in Wakwa (Ngaoundere Urbain HD) to 0.129 mf/ss in Nandeke (Meiganga HD). When assessed for participants aged more than 20 years, the CMFLs varied from 0 mf/ss in Mbakaou (Tibati HD) to 0.0716 mf/ss in Zouzami (Meiganga HD) (**Table 1**). At the level of the HD, the mf and nodule prevalences were estimated by grouping together the communities surveyed in the same HD. The prevalences recorded in 2020 in the surveyed HDs are shown in **Table 2**.

Within the HDs of the study region, nodules prevalences varied from 0.00 [0.00 - 0.77] in Meiganga HD to 2.36 [1.32 - 4.11] in Tibati HD. The mf prevalences varied from 0.00 [0.00 - 0.80] % in Banyo and Tignere HDs to 3.40 [2.16 - 5.23] in Meiganga HD. When HDs were grouped together, the nodule and mf prevalences recorded for the whole region were 0.87[0.64 - 1.19] % and 0.77 [0.54 - 1.07] % respectively (**Table 2**). **Fig 3** shows the mf and nodule prevalences reported in these HDs during the 1998-2002, 2010-2013 and 2020 surveys conducted in this region. Overall, the mf and nodule prevalences reported in the 2020 survey were significantly lower (p < 0.05) compared to those recorded in the 1998-2002 and 2010-2013 surveys (**Fig 3A and 3B**). Compared to the 2010-2013 study, a slight increase in mf prevalence was observed in Meiganga HD in 2020 (**Fig 3B**). At the community level, the baseline 1998-2002 mf prevalences ranging from 51.1% (Ngaoundal) to 77.1% (Ngaoundére Rural) have significantly decreased from 0% (Banyo or Meiganga) to 2.1% (Ngaoundal) in the 2020 follow-up survey (p < 0.05) (**Fig 4**).

**Table 1. Mf prevalences in the communities prospected during the 1998-2002 and 2020 epidemiological surveys conducted in Adamaoua Region.**

| Health District | Community | Geographic cordinates | | Baseline survey (1998-2002) | | | Follow up survey (2020) | | | |
| --- | --- | --- | --- | --- | --- | --- | --- | --- | --- | --- |
| | | Latitude | Longitude | # of persons examined | Mf prevalence (1998-2002) | # of persons examined | Mf preva-lence [95%IC] (2020) | p-value | CMFL (All) | CMFL (+20yrs) |
| Bankim | Mgbandji* | 5°59'22.5" | 11°16'36.5" | 30 | 80.2 | 87 | 1.14 [0.06 - 7.13] | < 5% | 0.023 | 0.035 |
| | Koumtchoum* | 6°26'81.8" | 11°12'14.8" | 30 | 38.9 | 227 | 0 | < 5% | – | – |
| | Moinkoing* | 6°02'58.3" | 11°24'40.2" | 32 | 55.7 | 100 | 0 | < 5% | – | – |
| Banyo | Mayo Dinga* | 6°25'18.2" | 11°34'15.0" | 30 | 60.3 | 173 | 0 | < 5% | – | – |
| | Mbamti* | 6°41'02.7" | 12°04'00.0" | 32 | 59.3 | 180 | 0 | < 5% | – | – |
| | Yimbere* | 6°27'35.3" | 11°34'269" | 30 | 50.9 | 239 | 0 | < 5% | – | – |
| Djohong | Gbatoua* | 6°50'42.6" | 14°42'56.4" | 33 | 65.7 | 146 | 0.68 [0.03 - 4.32] | < 5% | 0.008 | 0.013 |
| | Yamba* | 7°06'21.0" | 15°12'20.8" | 30 | 71.0 | 129 | 0 | < 5% | – | – |
| Meiganga | Korekoni* | 6°08'09.3" | 14°33'48.0" | 33 | 55.4 | 225 | 4.00 [1.96 - 7.70] | < 5% | 0.127 | 0.051 |
| | Lokoti | 6°22'00.0" | 14°20'00.0" | 43 | – | 89 | 1.12 [0.05 - 6.97] | – | 0.022 | 0.022 |
| | Zouzami* | 6°37'08.3" | 14°32'20.0" | 30 | 80.3 | 168 | 2.97 [1.10 - 7.17] | < 5% | 0.049 | 0.071 |
| | Nandeke* | 6°27'09.7" | 14°13'47.1" | 38 | 56.3 | 138 | 4.34 [1.77 - 9.63] | < 5% | 0.129 | 0.047 |
| Ngaoundal | Boy-Baya | 6°25'26.4" | 13°23'10.7" | 31 | – | 56 | 3.57 [0.62 - 13.38] | – | 0.048 | 0.048 |
| | Bagodo* | 6°25'26.4" | 13°23'10.7" | 50 | 55.6 | 110 | 0 | < 5% | – | – |
| | Ngaoundal* | 6°28'06.0" | 13°16'09.4" | 31 | 46.6 | 206 | 2.91 [1.18 - 6.52] | < 5% | 0.050 | 0.050 |
| Ngaoundere Rural | Nganha* | 7°26'08.8" | 13°55'43.2" | 30 | 80.2 | 169 | 0.59 [0.03 - 3.75] | < 5% | 0.010 | 0.010 |
| | Bakari Bata* | 6°55'41.9" | 14°35'51.7" | 33 | 59.1 | 213 | 0 | < 5% | – | – |
| | Berem* | 7°33'03.7" | 13°55'28.4" | 30 | 86.7 | 299 | 0 | < 5% | – | – |
| | Nyassar* | 7°31'40.1" | 14°01"43.7" | 30 | 82.5 | 281 | 0 | < 5% | – | – |
| Ngaoundere Urbain | Wakwa* | 7°16'05.8" | 13°33'01.6" | 30 | 66.0 | 95 | 4.21 [1.35 - 11.03] | < 5% | 0.026 | 0.051 |
| | Gadamabanga | 7°21'37.7" | 13°35'05.5" | 32 | – | 78 | 0 | – | – | – |
| | Mballang | 7°18'00.8" | 13°44'30.2" | 30 | – | 110 | 0 | – | – | – |
| Tibati | Mbakaou* | 6°18'13.9" | 12°47'48,0" | 52 | 67.4 | 197 | 0.50 [0.02 - 3.23] | < 5% | 0.006 | 0 |
| | Alhamissa* | 6°30'01,9" | 12°41'58.9" | 30 | 36.7 | 113 | 0 | < 5% | – | – |
| | Djombi | 6°41'28.8" | 12°36'18.4" | 33 | – | 240 | 0 | – | – | – |
| Tignere | Libong | 7°23'21.5" | 13°00'06.4" | 30 | – | 171 | 0 | – | – | – |
| | Mayo-Djarandi | 7°22'05.0" | 12°38'59.3" | 30 | – | 175 | 0 | – | – | – |
| | Mayo-Kaloua* | 7°05'13.7" | 12°28'15.0" | 23 | 74.1 | 187 | 0 | < 5% | – | – |
| | Djamtari* | 7°42'03.4" | 12°15'01.9" | 36 | 87.7 | 213 | 0 | < 5% | – | – |

*Communty prospected in 1998-2002 and 2020; **#**: number; **Mf**: microfilaria; -: No data available.

The map was created using QGIS version 3.10.6. The "Link to the base layer is: ptnd. 0004224.s004; *www.mapcruzin.com* (World, Cameroon, Administrative division, hydrograpgy, roads and S4 Cameroon health districts shapefiles). The shapefile source is: Sub Department for Epidemiological Surveillance in the Ministry of Public Health Cameroon (Free GIS software (Projects, Shapefiles, maps, etc…)". This map was made by Houyamné Gong-Gali Adam Byang from the Department of Animal Biology of the University of Ngaounderé, Ngaoundéré, Cameroon.

## Trend in ivermectin treatment and *O. volvulus* infection distribution

Of the 4,814 participants interviewed for ivermectin treatment, ≈ 28% (N = 1,351) representing almost one third of the studied population said they had never been treated, while ≈ 72% (N = 3,463) declared having received ivermectin treatment at least once in the past 6 years (2014-2019). The proportion of participants who had never been treated with ivermectin in

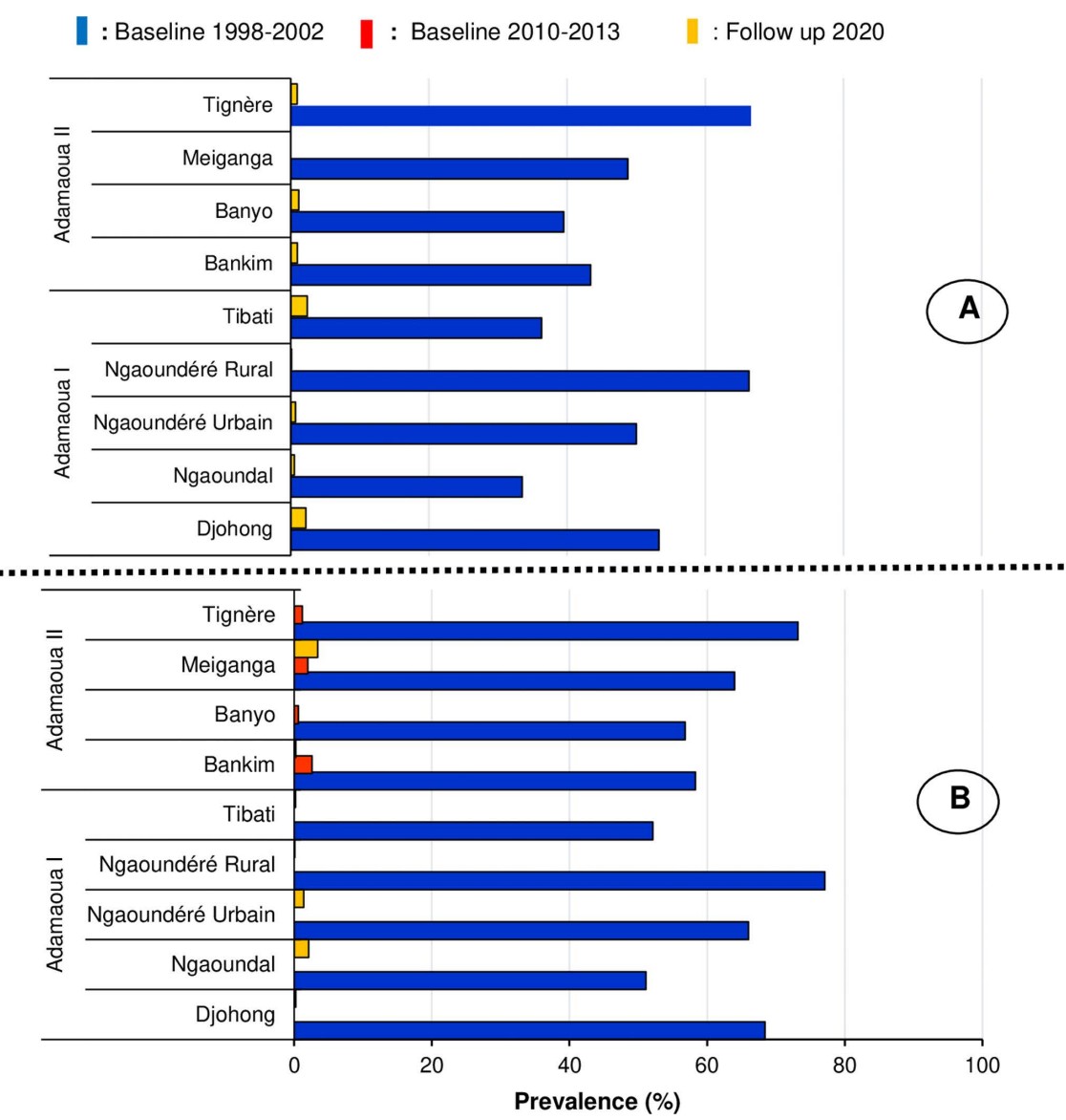

**Fig 3. Nodule (A) and microfilaria (B) prevalences recorded in the surveyed Health Districts.**

the surveyed communities ranged from 1% (Djamtari, Tignere HD) to 11% (Mbamti, Banyo HD). Among those who were treated, 86% (N = 2,977) received the last treatment in 2019, 9.7% (N = 336) in 2018, 4% (N = 141) in 2017 and the remaining 0.3% (N= 9) in 2016, 2015 and 2014. Of the 4,814 people examined, 37 were tested positive for *O. volvulus* infection with 25 belonging to the treated group, i.e., 25/3463 (0.72%) consisting of individuals who have received at least one ivermectin treatment, and 12 belonging to the untreated group, i.e., 12/1351 (0.88%) made up with ivermectin non-compliers. The mf prevalences in treated and untreated individuals were not statistically different ($\chi^2$ = 0.3525, p = 0.5527).

The therapeutic coverage rates (TCRs) reported and those surveyed in 2019 in the HDs of Adamaoua Region are shown in **Fig 5**. The TCRs reported by the National Onchocerciasis Control Programme (NOCP) ranged from 78.8% (Tibati HD) to 83.1% (Djohong HD), while those surveyed ranged from 41.3% (Ngaoundere Urbain HD) to 77.5% (Tibati HD). Overall,

Table 2. Microfilaria and nodule prevalences recorded in the health districts of the Adamaoua Region.

| Health District | # of persons examined | Prevalence | |
|---|---|---|---|
| | | Nodule | Mf |
| Bankim | 414 | 0.96 [0.31- 2.63] | 0.24 [0.01 - 1.55] |
| Banyo | 592 | 1.18 [0.52 - 2.53] | 0.00 [0.00 - 0.80] |
| Djohong | 275 | 2.18 [0.89 - 4.92] | 0.36 [0.02 - 2.33] |
| Meiganga | 620 | 0.00 [0.00 - 0.77] | 3.40 [2.16 - 5.23] |
| Ngaoundal | 372 | 0.53 [0.09 - 2.14] | 2.15 [1.00 - 4.36] |
| Ngaoundere rural | 962 | 0.10 [0.00 - 0.67] | 0.10 [0.00 - 0.67] |
| Ngaoundere urbain | 283 | 0.71 [0.12 - 2.81] | 1.41 [0.45 - 3.82] |
| Tibati | 550 | 2.36 [1.32 - 4.11] | 0.18 [0.01 - 1.17] |
| Tignere | 764 | 0.94 [0.41 - 2.01] | 0.00 [0.00 - 0.64] |
| **TOTAL** | **4,814** | **0.87 [0.64 - 1.19]** | 0.77 [0.54 - 1.07] |

Mf: microfilaria

the TCRs reported by the NOCP were significantly higher than those surveyed, with an exception in Tibati HD, where the reported coverage rate was similar to that surveyed (**Fig 5**).

In this study, the adherence rates of population to ivermectin treatment (i.e., the percentage of people who took ivermectin among the eligible population in the community or HD) by gender and age group were also investigated (**Fig 6**). At the regional level, men and women showed comparable adherence rates during the 2019 ivermectin mass distribution campaign, i.e., 48.1% (N = 1,433) and 51.9% (N = 1,544) respectively (p > 0.05%). This equality in the treatment adherence was also noted at the HD level, with however an exception in Tibati and Tignere HDs, where women's adherence was significantly higher than men's, while this trend was reversed in Tibati and Tignere HDs (**Fig 6**).

In general, both men women showed adherence rates < 65%. When HDs were grouped together, the population adherence rate by age group was shown in **Fig 7**. Overall, all age groups showed adherence rates ≤ 65%, with however an exception in the 40-50 age group where the adherence rate was ≈ 70%.

## Discussion

The objective of this study was to assess the status of onchocerciasis transmission in the Adamaoua Region after two decades of mass treatment with ivermectin. Overall, the nodule and microfilaria prevalences recorded during this 2020 epidemiological study in the surveyed communities and HDs of the region were below 5% with most of them exhibiting zero mf prevalence. Based on the criteria applied for delineating onchocerciasis foci into endemicity levels, all these communities and HDs may be currently classified as sporadic [49] or hypo-endemic [50,51] for the disease. The CMFLs assessed according to Remme and Colleagues [52] showed less than 1 mf/ss indicating low parasitic loads in human population, with however an important proportion of parasites circulating in the youngest population. The parasitic loads assessed in the surveyed communities were significantly low compared to those reported in the same communities during the 1998-2002 REMO data collection conducted in Cameroon [53,54]. This decline in onchocerciasis intensity has previously been reported in Adamaoua II during the 2013 epidemiological assessments conducted in Cameroon as part of the impact assessment of the CDTI strategy in countries of African Programme for Onchocerciasis Control [37]. The low prevalences of the disease recorded in the surveyed communities is attributed to efforts put in place in controlling the disease through an effective involvement

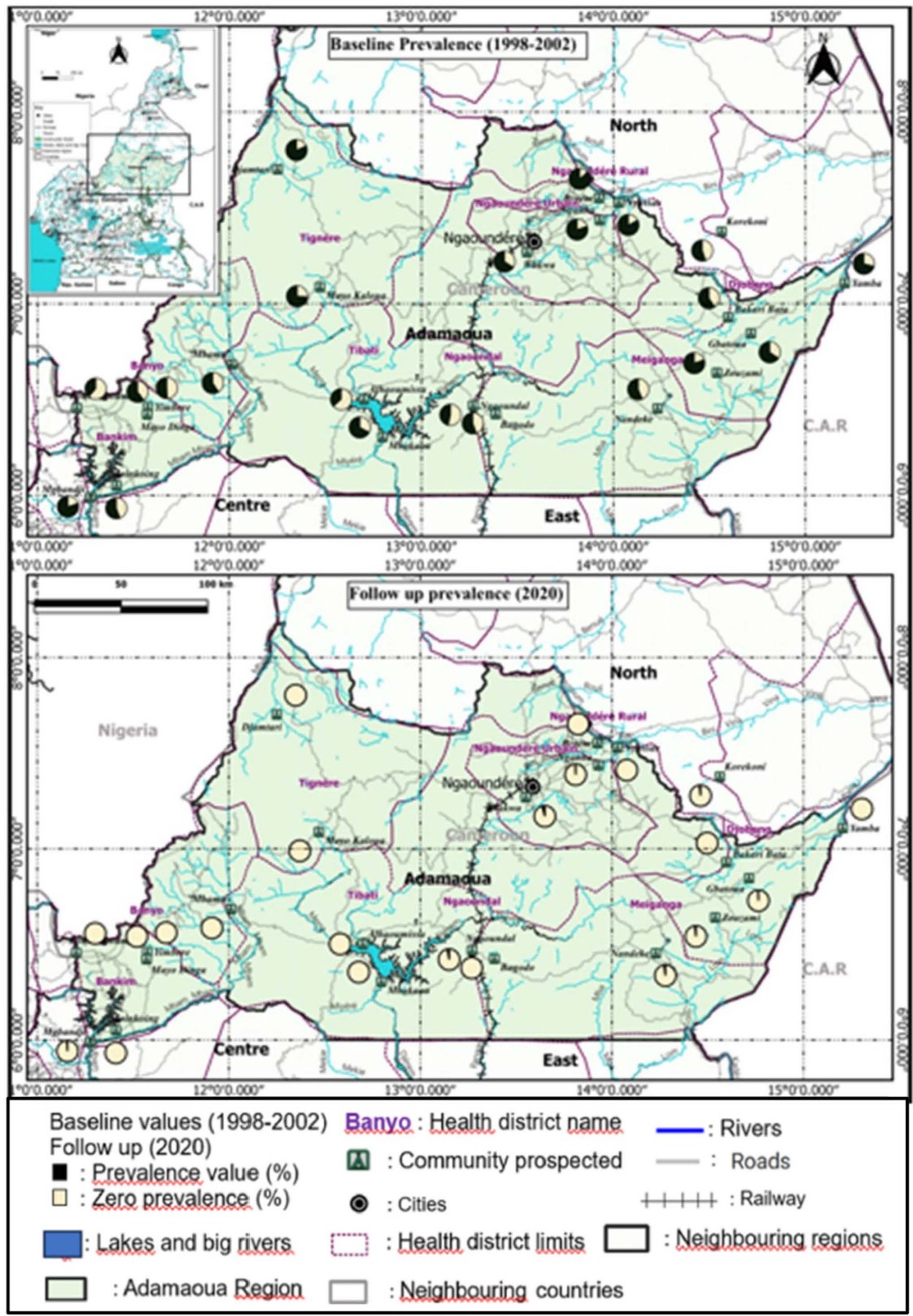

**Fig 4. Trend in microfilaria prevalence in 22 communities surveyed in the 1998-2002 baseline and the 2020 follow-up surveys in the Adamaoua Region.**

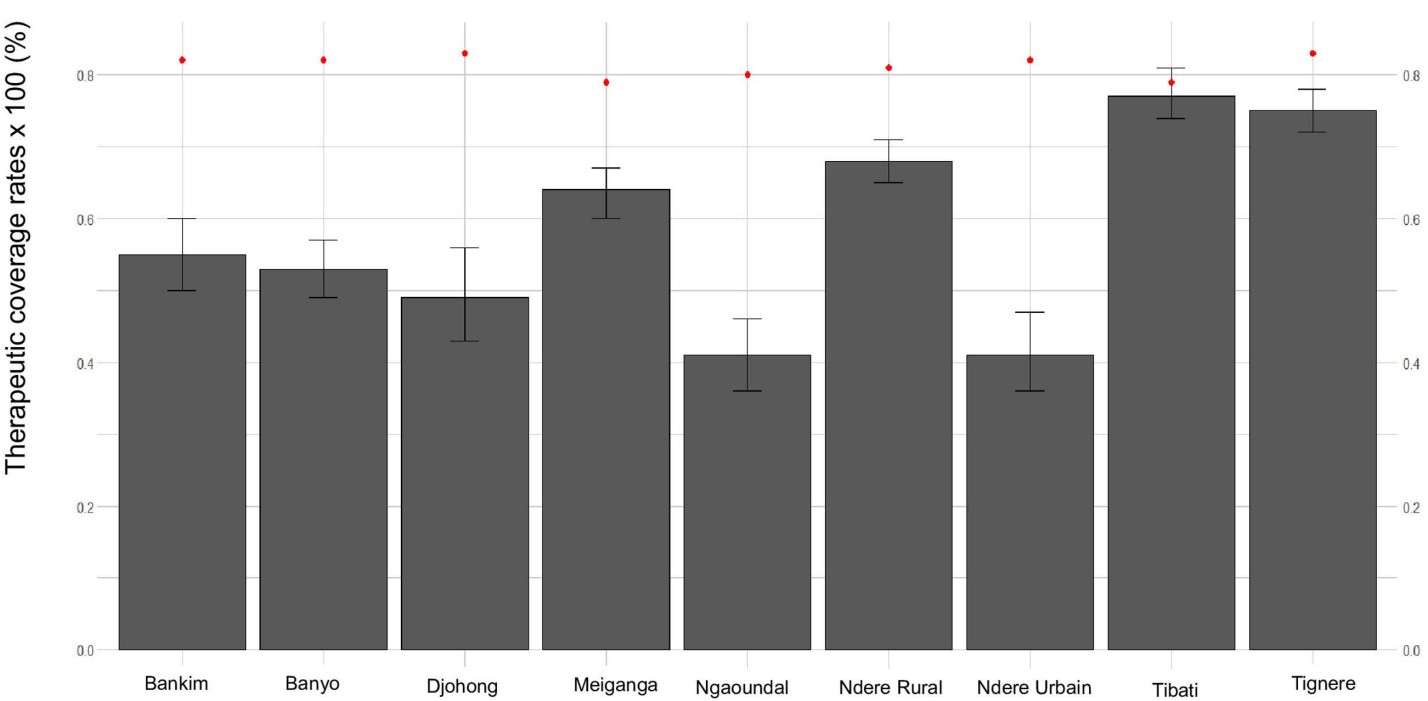

**Fig 5. Therapeutic coverage rates reported (Red dot) and surveyed (histogram bar) in 2019 in the health districts of Adamaoua Region.**

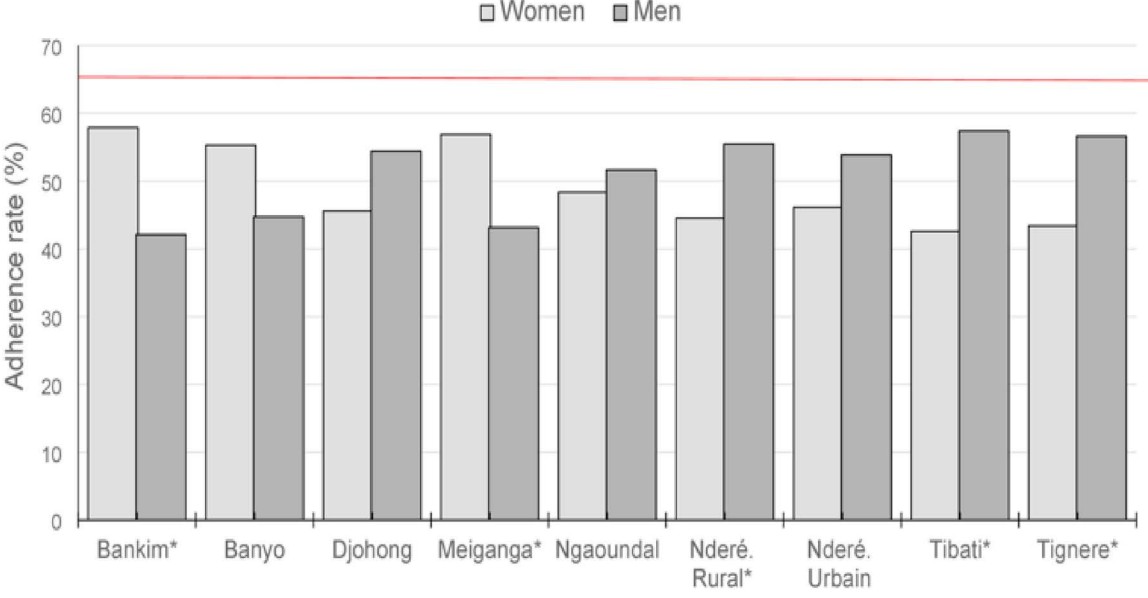

**Fig 6. Population adherence to ivermectin treatment rates by gender during the 2019 mass distribution in the 9 health districts of Adamaoua Region.**

of communities in ivermectin mass treatment during the past 20 consecutive years. In the context of achieving the elimination of onchocerciasis by 2030, data presented in this study are promising compared to the findings recently reported in other regions of Cameroon [55–57].

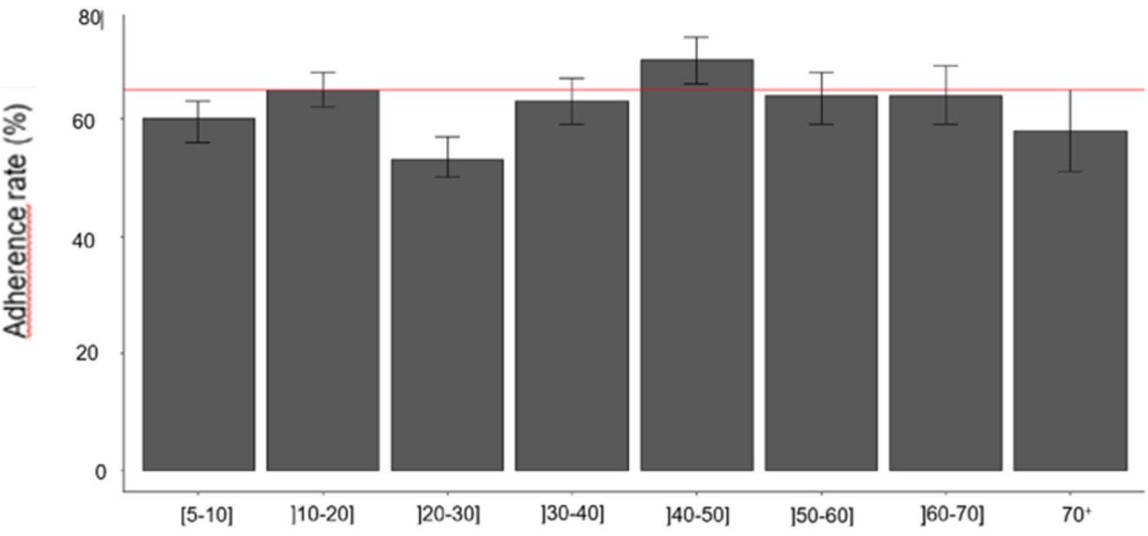

**Fig 7. Population adherence rates according to age groups during the 2019 ivermectin mass treatment in the 9 health districts of Adamaoua Region.**

The slight increase in mf prevalence noted in Meiganga HD during this 2020 epidemiological study compared to that of 2010-2013 may be attributed to a decline in the population's adherence in ivermectin treatment. This confirms the low adherence (< 65%) of population in ivermectin treatment noted in most of the surveyed HDs and all age groups of the eligible population. Also, the existence in the prospected communities of systematic non-compliers to ivermectin representing ≈ 4% to 11% in each surveyed community and accounting for half of the total infected individuals identified may favour the observed increase in disease prevalence in some HDs. It must be underline that, non-compliers to ivermectin treatment represent microfilaria reservoirs for the human population, and this may induce active transmission in this environment where microfilaria parasites and blackflies co-exist.

Indeed, infective stage ($L_3$) of these microfilariae are picked up from these non-compliers by *Simulium* blackflies during their blood meals and spread to individuals free of onchocerciasis. The permanent feeding behaviour of blackflies favours the spread of parasites from one person to another, infesting several human communities with onchocerciasis. In the savannah zones such as the Adamaoua Region, the spread of these parasitic forms is easier, with the passive movement of blackflies that can reach locations situated tens or even hundreds of kilometers away. Regarding parasite transmission, *Simulium damnosum* s.s, *S. sirbanum* and *S. squamosum* are known as major vectors of onchocerciasis in the Adamaoua Region [58]. Although the presence of these vectors in the region needs to be confirmed through an updated taxonomic study on *Simulium* blackflies, we might highlight that these blackfly species exhibit the phenomenon of "limitation" known as one of the factors favoring the transmission of the disease [59,60]. The phenomenon is described as the situation where vectors are efficient even at very low parasite densities [61]. The low intensity of the disease expressed by CMFLs (< 1mf/ss) and the limited number of infected individuals reported in this study suggest ongoing transmission of the parasite in this region but at low level. Although the decrease in the prevalence of onchocerciasis is noted in the surveyed communities of the region, it should be mentioned that, the local *O. volvulus* parasite population has not yet been reduced to a level below the transmission breakpoint. Data collected during 2010-2013 and 2020 studies from Meiganga reflect this situation and suggest that the transmission of *O.*

*volvulus* infection in this HD as in others may evolve in a piecemeal fashion, indicating that if no special attention is given in delivering ivermectin to population, a recrudescence of the disease may occur rapidly in this region.

Although the entire Adamaoua Region is under ivermectin treatment, the surveyed therapeutic coverage rates in 2019 reveal that 5/9 of the HDs are far below the minimum rate of 65% recommended by the former APOC in the framework of onchocerciasis control as a public health problem. Overall, the surveyed coverage rates recorded are lower than those reported by the NOCP in 2019. Although resulting from oral statements of participants, these surveyed coverage rates may be considered as more informative and reflect the status in ivermectin mass treatment in the Adamaoua Region.

The reported and surveyed coverage rates in each HD of Adamaoua Region were only statistically comparable for the Tibati HD, indicating that the CDTI data provided by the local community distributors of ivermectin and health system are in agreement with the participants' statements. Furthermore, the two coverage rates are almost all equal of the desired therapeutic coverage (80%) of the eligible population as per the WHO recommendation. According to the WHO, this observation confers a validation of the CDTI strategy in this HD [62]. In the other HDs, the therapeutic coverage rates reported by NOCP in 2019 would have been for the most part overestimated at different levels of the recording system used in documenting ivermectin treatment. The possible reasons for this overestimation of coverage rates may be as follows: (1) the justification of the ability to carry out the task, (2) the renewal of the confidence of the hierarchy and (3) the hope of obtaining a financial or material compensation. The low surveyed therapeutic coverage rates reported in 2019 in most of the HDs confirms the poor adherence of the eligible population in ivermectin treatment. This observation is a key information to consider for the elimination of the onchocerciasis in Adamaoua Region. Such a situation calls for additional efforts in delivering ivermectin in the region through increased sensitization of local populations. However, combining the ivermectin treatment with complementary strategies such vector control, enhanced CDTI, community-directed treatment (CDT) with drug combinations or new drugs, and test-and-treat (TNT) [63], may accelerate the stop of *O. volvulus* transmission in this region.

The data reported in this study are close to those indicating the elimination threshold as previously recommended by the former APOC [64] (1) less than 5% prevalence in all surveyed communities and (2) less than 1% in 90% of surveyed communities. All the 29 communities surveyed had mf prevalence below 5% and 70% had mf prevalence below 1%. The current profile of the disease indicates low mf prevalences (< 5%) and CMFLs (< 1mf/ss) across the surveyed communities of the Adamaoua Region. These data are in agreement with WHO guidelines [65], confirming that onchocerciasis is no longer a public health problem in this region. However, to better characterize the profile of the disease in Adamaoua Region, it would be advisable to (1) carry out investigations in several other communities not included in this study, (2) increase the size of the study population and (3) and above all to carry out OV16 tests to demonstrate children's exposure to the infection, as recommended by the WHO. Onchocerciasis is endemic in all regions of Cameroon and its elimination should start in one region and spread to others. The data presented in this study indicate that the transmission of *O. volvulus* can be interrupted in the short term in the Adamaoua Region. Indeed, the Adamaoua Region is known as the "Watershep of Cameroon", where many rivers especially those infested by blackflies, have their source. Starting the interruption of the disease transmission in this region and extending it to others would be a step towards ridding the country of *O. volvulus* infection. The Adamaoua Region is geographically the transition zone between the southern and northern regions of Cameroon, i.e., the southern forest and the northern savannah areas. The fight against *O. volvulus*

infection in Adamaoua-North and Adamaoua-South directions could be an effective operational intervention plan to be adopted for the future elimination of onchocerciasis in Cameroon.

## Conclusion

The results of this study show a drastic decline in the prevalence of onchocerciasis in the Adamaoua Region after 20 consecutive years of CDTI. The mf prevalences recorded indicate that, the disease has reached relatively low levels of endemicity in the surveyed communities suggesting that the interruption of its transmission may be possible in the near future in this part of the country. However, further additional efforts are expected for the population adherence in ivermectin mass treatment to move towards elimination. This goal may be achieved in a very short term between 2025 and 2030 by improving the annual treatment of populations with ivermectin at > 85% therapeutic coverage rates. Furthermore, complementary strategies including more in-depth sensitization of community members targeting non-compliers to ivermectin treatment, physical destruction of *Simulium* breeding sites through community-directed approach ("Slash and Clear") would accelerate the stop of the disease transmission in this part of the country. These data are of great importance to the Ministry of Public Health, in particularly its NOCP, and could serve as a compass for the elimination of onchocerciasis in Cameroon and Central Africa.

## Supporting information

**S1 File. Data on nodule and mf prevalences collected in 1998-2001, 2010-2013 and 2019-2020.**
(DOCX)

**S2 File. Geographic cordinates and mf prevalences recorded 1998-2001 to 2020 in 24 communities.**
(DOCX)

**S3 File. Therapeutic coverages reported by the NOCP from 2014 to 2019 and surveyed coverages (2020).**
(DOCX)

**S4 File. Raw data in adherence for ivermection treatment by gender in 2019.**
(DOCX)

**S5 File. Percentage of people treated in the age groups of the study population.**
(DOCX)

**S6 File. 2019-2020 survey database.**
(XLSX)

## Acknowledgments

The authors are grateful to the Regional Health Delegate of Adamaoua Region, all health staff at the levels of the health districts and health areas, the traditional chiefs and their respective populations for their active involvement for data collection.

## Author contributions

**Conceptualization:** Philippe Bienvenu Nwane, Hugues Clotaire Nana-Djeunga, Patrice Nkwelle, Joseph Kamgno.

**Data curation:** Philippe Bienvenu Nwane, Fesuh Nono Bertrand.

**Formal analysis:** Philippe Bienvenu Nwane, Hugues Clotaire Nana-Djeunga, Fesuh Nono Bertrand, Yannick Emalio Niamsi.

**Investigation:** Philippe Bienvenu Nwane, Narcisse Nzune Toche, André Domché, Guy Roger Njitchuang, Martine Augusta Flore Tsassé, Jean Bopda, Steve Mbickmen, Aubin Balog, Alexis Nkouelle, Clarisse Ebene, Honoré Obama, Paul Messi, Benjamin Biholong.

**Methodology:** Philippe Bienvenu Nwane, Martine Augusta Flore Tsassé, Jean Bopda, Steve Mbickmen, Aubin Balog.

**Supervision:** Philippe Bienvenu Nwane, Hugues Clotaire Nana-Djeunga.

**Validation:** Philippe Bienvenu Nwane, Fesuh Nono Bertrand.

**Visualization:** Serge Billong, Georges Nko'o Ayissi.

**Writing – original draft:** Philippe Bienvenu Nwane.

**Writing – review & editing:** Philippe Bienvenu Nwane, André Domché, Joseph Kamgno.

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
