## [Decision Letter · Decision Letter 0]

4 Oct 2023

Dear Dr Nwane,

Thank you very much for submitting your manuscript "Elimination of Onchocerciasis in the Adamaoua region of Cameroon: A big step towards the end." for consideration at PLOS Neglected Tropical Diseases. As with all papers reviewed by the journal, your manuscript was reviewed by members of the editorial board and by several independent reviewers. In light of the reviews (below this email), we would like to invite the resubmission of a significantly-revised version that takes into account the reviewers' comments. 

We cannot make any decision about publication until we have seen the revised manuscript and your response to the reviewers' comments. Your revised manuscript is also likely to be sent to reviewers for further evaluation.

Sincerely,

Uwem Friday Ekpo, PhD

Academic Editor

Dileepa Ediriweera

Section Editor

Reviewer's Responses to Questions

**Key Review Criteria Required for Acceptance?**

**Methods**

-Are the objectives of the study clearly articulated with a clear testable hypothesis stated?

-Is the study design appropriate to address the stated objectives?

-Is the population clearly described and appropriate for the hypothesis being tested?

-Is the sample size sufficient to ensure adequate power to address the hypothesis being tested?

-Were correct statistical analysis used to support conclusions?

-Are there concerns about ethical or regulatory requirements being met?

Reviewer #1: The objectives are clearly articular; the study design is appropriate. However, sampling and methodology and limitations at different point are not clearly presented limitations. Also, statistically analysis to make the conclusion appear inadequate

Reviewer #2: The objectives of the study are clearly articulated, with a clear testable hypothesis

- the study design is not appropriate to address the stated objectives.e.g the study is impact of control strategy on oncherciasis transmission and not on elimination as the title suggests.

- the population is clearly described and very appropriate for the hypothesis being tested

- the sample size is quite adequate to address the hypothesis being tested

- correct statistical methods were adopted for data analysis and support for conclusion

- ethical issues concerning the use of corneo- scleral punch in oncherciasis diagnosis was not addressed.

**Results**

-Does the analysis presented match the analysis plan?

-Are the results clearly and completely presented?

-Are the figures (Tables, Images) of sufficient quality for clarity?

Reviewer #1: Results need to been well articular towards objectives. As mentioned earlier, statistical analysis could be done better

Reviewer #2: The analysis presented were in tandem with analysis plan.The authors did not present the pre.control data of the study communities to enable for proper analysis of the control strategy through CDTI this is another drawback.

The results were clearly but not completely presented 

- the figures, tables and images are not sufficient to represent the quantity collected in the study.

**Conclusions**

-Are the conclusions supported by the data presented?

-Are the limitations of analysis clearly described?

-Do the authors discuss how these data can be helpful to advance our understanding of the topic under study?

-Is public health relevance addressed?

Reviewer #1: Conclusion appear a bit premature without indebt consideration of sampling, analysis and diseases epidemiology. Limitation of sampling, analysis and disease epidemiology need to be considered in analysis, discussion and conclusion

Reviewer #2: The conclusion are fairly supported by the data presented

- the limitations of the study are stated in the manuscript.e.g not using ov16 Elisa to test children in the quest for elimination 

- the authors stated how the results of this can be utilized for policy formulation in the control and elimination of oncherciasis

- the public health relevance of the study is adequately stated.

**Editorial and Data Presentation Modifications?**

Reviewer #1: minor revision

Reviewer #2: The topic should be changed to read ' status of human oncherciasis transmission in the Adamaoua region of Cameroon after 20 years of ivermectin control ' this is due to the fact that the study is assessing impact of ivermectin control and not assessing elimination

**Summary and General Comments**

Reviewer #1: This is a important paper that attend to demonstrate progress to elimination of onchocerciasis. Findings look interesting but for grounded results to support the conclusion, limitations and other factors such environment and treatment need to be considered. This paper will benefit from major revision as well as proof reading for English language use

Reviewer #2: The manuscript is well written but needs some editing because of numerous typographical errors

- the major drawback is that the authors address ethical issues concerning the use of corneo-scleral punch in the diagnosis of oncherciasis

- policy makers involved in oncherciasis transmission will find the outcome of the study very useful

- the project was well executed

PLOS authors have the option to publish the peer review history of their article (what does this mean? ). If published, this will include your full peer review and any attached files.

**Do you want your identity to be public for this peer review?** For information about this choice, including consent withdrawal, please see our Privacy Policy .

Reviewer #1: No

Reviewer #2: No
---

## [Decision Letter · Decision Letter 1]

11 Jun 2024

Dear Dr Nwane,

Thank you very much for submitting your manuscript "Status of human oncherciasis transmission in the Adamaoua region of Cameroon after 20 years of ivermectin mass distribution" for consideration at PLOS Neglected Tropical Diseases. As with all papers reviewed by the journal, your manuscript was reviewed by members of the editorial board and by several independent reviewers. In light of the reviews (below this email), we would like to invite the resubmission of a significantly-revised version that takes into account the reviewers' comments. Please address the comments of Reviewer #1 in particular.

We cannot make any decision about publication until we have seen the revised manuscript and your response to the reviewers' comments. Your revised manuscript is also likely to be sent to reviewers for further evaluation.

Sincerely,

Uwem Friday Ekpo, PhD

Academic Editor

Dileepa Ediriweera

Section Editor

Please attain the comments of Reviewer #1

Reviewer's Responses to Questions

**Key Review Criteria Required for Acceptance?**

**Methods**

-Are the objectives of the study clearly articulated with a clear testable hypothesis stated?

-Is the study design appropriate to address the stated objectives?

-Is the population clearly described and appropriate for the hypothesis being tested?

-Is the sample size sufficient to ensure adequate power to address the hypothesis being tested?

-Were correct statistical analysis used to support conclusions?

-Are there concerns about ethical or regulatory requirements being met?

Reviewer #1: Statistics is still lacking. For example the authors mention ANOVA and Welch back in abstract and these do not appear anywhere else. Only P<values provided without the statistics etc. See annotated manuscript. Background information such as treatment performance during the past 20 years need to be highlighted. Onchocerciasis study revolve around river, blackfly and human being. The communities selected are not situated with respect to river and thus flies - were they first line villages or second line? any update of breeding sites

Reviewer #2: The title is appropriate

The objectives are we spelt out

The study design and population of study are quite adequate.

The ethical concerns have been addressed

Results

-Does the analysis presented match the analysis plan?

-Are the results clearly and completely presented?

-Are the figures (Tables, Images) of sufficient quality for clarity?

Reviewer #1: Yes, the table need a bit tidying up. see annotated manuscript

Reviewer #2: The analysis and result presented are clearly stated

**Conclusions**

-Are the conclusions supported by the data presented?

-Are the limitations of analysis clearly described?

-Do the authors discuss how these data can be helpful to advance our understanding of the topic under study?

-Is public health relevance addressed?

Reviewer #1: Yes, they are supported by data but much work need to be done to clearly present the findings; limitation not clear discuss. See annotated manuscript

Reviewer #2: The conclusion supports the data presented

The public health relevance of the study is adequately discussed.

**Editorial and Data Presentation Modifications?**

Reviewer #1: The English/language use is poor. This is a major issue throughout the paper. The authors should seek help.

Reviewer #2: The authors have modified the manuscript in line with the comments and suggestions.

It can now be accepted for publication.

**Summary and General Comments**

Reviewer #1: General feedback

1. language is very poor. Please, see annotated manuscripts

2. Background should bring out treatment history, coverage evaluation 

3. Statistics not adequate - e.g sampling and inferential statistics

Reviewer #2: This study is very valuable to policy makers involved onchocerciasis elimination.There might be need to compliment the ongoing ivermectin treatment with vector control.

PLOS authors have the option to publish the peer review history of their article (what does this mean? ). If published, this will include your full peer review and any attached files.

**Do you want your identity to be public for this peer review?** For information about this choice, including consent withdrawal, please see our Privacy Policy .

Reviewer #1: No

Reviewer #2: No
---

## [Decision Letter · Decision Letter 2]

16 Sep 2024

Dear Dr Nwane,

Thank you very much for submitting your manuscript "Status of human onchocerciasis transmission in the Adamaoua region of Cameroon after 20 years of ivermectin mass distribution" for consideration at PLOS Neglected Tropical Diseases. As with all papers reviewed by the journal, your manuscript was reviewed by members of the editorial board and by several independent reviewers. The reviewers appreciated the attention to an important topic. Based on the reviews, we are likely to accept this manuscript for publication, providing that you modify the manuscript according to the review recommendations. 

Sincerely,

Uwem Friday Ekpo, PhD

Academic Editor

Dileepa Ediriweera

Section Editor

Reviewer's Responses to Questions

**Key Review Criteria Required for Acceptance?**

**Methods**

-Are the objectives of the study clearly articulated with a clear testable hypothesis stated?

-Is the study design appropriate to address the stated objectives?

-Is the population clearly described and appropriate for the hypothesis being tested?

-Is the sample size sufficient to ensure adequate power to address the hypothesis being tested?

-Were correct statistical analysis used to support conclusions?

-Are there concerns about ethical or regulatory requirements being met?

Reviewer #1: (No Response)

Reviewer #2: -the objectives o the study are clearly stated with testable hypothesis

-the study designs conforms with standard protocol

-the population were adequately described to justify the study

-the sample size is sufficient to derive good inference

-statistical analysis are appropriate and well carried out

-ethical concerns have been addressed

**Results**

-Does the analysis presented match the analysis plan?

-Are the results clearly and completely presented?

-Are the figures (Tables, Images) of sufficient quality for clarity?

Reviewer #1: (No Response)

Reviewer #2: The results are clearly presented.

- the figures and tables are of sufficient quality

and justify the analyzed data

**Conclusions**

-Are the conclusions supported by the data presented?

-Are the limitations of analysis clearly described?

-Do the authors discuss how these data can be helpful to advance our understanding of the topic under study?

-Is public health relevance addressed?

Reviewer #1: There is much improvement and clarify. However, some aspects mentioned earlier have not be considered. e.g the mention of ANOVA only in abstract; approach to elimination by Cameroon - Region or transmission zone. It will appear it is by Region as per author's discussion. But I think the Cameroon approach is not very clear or stated as yet. Still some minor English to Correct eg first sentence of conclusion in the abstract start in small letter.

Reviewer #2: The manuscript is well concluded and supported with data, the authors have stated their limitations. The implication of their findings for policy makers in onchocerciasis control and elimination are clearly stated

**Editorial and Data Presentation Modifications?**

Reviewer #1: (No Response)

Reviewer #2: none

**Summary and General Comments**

Reviewer #1: There is much improvement and clarify. However, some aspects mentioned earlier have not be considered. e.g the mention of ANOVA only in abstract; approach to elimination by Cameroon - Region or transmission zone. It will appear it is by Region as per author's discussion. But I think the Cameroon approach is not very clear or stated as yet. Still some minor English to Correct eg first sentence of conclusion in the abstract start in small letter.

Reviewer #2: The manuscript has addressed a major finding in the control efforts the government.There is need for the government to intensify efforts in their treatment and geographical coverage to achieve better results.

PLOS authors have the option to publish the peer review history of their article (what does this mean? ). If published, this will include your full peer review and any attached files.

**Do you want your identity to be public for this peer review?** For information about this choice, including consent withdrawal, please see our Privacy Policy .

Reviewer #1: No

Reviewer #2: No

Figure Files:

Data Requirements:

Reproducibility:

References

---

## [Editor Report · Decision Letter 3]

16 Dec 2024

Dear Dr Nwane,

We are pleased to inform you that your manuscript 'Status of human onchocerciasis transmission in the Adamaoua region of Cameroon after 20 years of ivermectin mass distribution' has been provisionally accepted for publication in PLOS Neglected Tropical Diseases.

Best regards,

Uwem Friday Ekpo, PhD

Academic Editor

Dileepa Ediriweera

Section Editor

Shaden Kamhawi

co-Editor-in-Chief

Paul Brindley

co-Editor-in-Chief

---

## [Editor Report · Acceptance letter]

Dear Dr Nwane,

We are delighted to inform you that your manuscript, " Status of human onchocerciasis transmission in the Adamaoua Region of Cameroon after 20 years of ivermectin mass distribution ," has been formally accepted for publication in PLOS Neglected Tropical Diseases.

Best regards,

Shaden Kamhawi

co-Editor-in-Chief

Paul Brindley

co-Editor-in-Chief
